# Some Important Metabolites Produced by Lactic Acid Bacteria Originated from Kimchi

**DOI:** 10.3390/foods10092148

**Published:** 2021-09-10

**Authors:** Se-Jin Lee, Hye-Sung Jeon, Ji-Yeon Yoo, Jeong-Hwan Kim

**Affiliations:** 1Division of Applied Life Science (BK21 Four), Graduate School, Gyeongsang National University, Jinju 52828, Korea; tpwls5151@naver.com (S.-J.L.); hye4098@naver.com (H.-S.J.); u6183@naver.com (J.-Y.Y.); 2Institute of Agriculture and Life Science, Gyeongsang National University, Jinju 52828, Korea

**Keywords:** kimchi, kimchi LAB, metabolites, bacteriocins, GABA, ornithine, mannitol, exopolysaccharides, 2-hydroxyisocaproic acid, 3-phenyllactic acid

## Abstract

Lactic acid bacteria (LAB) have been used for various food fermentations for thousands of years. Recently, LAB are receiving increased attention due to their great potential as probiotics for man and animals, and also as cell factories for producing enzymes, antibodies, vitamins, exopolysaccharides, and various feedstocks. LAB are safe organisms with GRAS (generally recognized as safe) status and possess relatively simple metabolic pathways easily subjected to modifications. However, relatively few studies have been carried out on LAB inhabiting plants compared to dairy LAB. Kimchi is a Korean traditional fermented vegetable, and its fermentation is carried out by LAB inhabiting plant raw materials of kimchi. Kimchi represents a model food with low pH and is fermented at low temperatures and in anaerobic environments. LAB have been adjusting to kimchi environments, and produce various metabolites such as bacteriocins, γ-aminobutyric acid, ornithine, exopolysaccharides, mannitol, etc. as products of metabolic efforts to adjust to the environments. The metabolites also contribute to the known health-promoting effects of kimchi. Due to the recent progress in multi-omics technologies, identification of genes and gene products responsible for the synthesis of functional metabolites becomes easier than before. With the aid of tools of metabolic engineering and synthetic biology, it can be envisioned that LAB strains producing valuable metabolites in large quantities will be constructed and used as starters for foods and probiotics for improving human health. Such LAB strains can also be useful as production hosts for value-added products for food, feed, and pharmaceutical industries. In this review, recent findings on the selected metabolites produced by kimchi LAB are discussed, and the potentials of metabolites will be mentioned.

## 1. Introduction

Kimchi is a traditional Korean vegetable food produced through fermentation carried out by lactic acid bacteria (LAB) originated from the raw ingredients of kimchi [1,2]. The major ingredients of kimchi are baechu (kimchi cabbage), radish, cucumber, and other vegetables, and baechu kimchi is the most common and popular type of kimchi in Korea. Other minor ingredients are added as seasoning, and they include chives, garlic, red pepper powder, ginger, leek, green onion, salt, and jeotgal (salted and fermented seafood) or aekjeot (liquid-type jeotgal, similar to fish sauce). The minor ingredients are first mixed together, and then added to baechu previously salted in brine overnight and washed with tap water. Depending on provinces and personal preferences, other vegetables or fruits such as pear are added [3]. A typical composition of baechu kimchi is as follows: the salted bachu (100%) is mixed with radish (13%), red pepper powder (3.5%), garlic (1.4%), ginger (0.6%), aekjeot (2.2%), sugar (1.0%), and green onion (2%) [3]. The final salt level is around 2.5%. Various biofunctional compounds are present in kimchi, and they are derived from either the raw materials themselves or the metabolic activities of LAB during fermentation. Kimchi fermentation is started by LAB originated from raw ingredients, and garlic is the most important source [2]. Ginger and leek are also important sources, whereas baechu contains a low number of LAB [2]. Fermentation proceeds at a refrigerating temp. between −1 °C and 7 °C for several months. During fermentation, various LAB species proliferate in sequence and produce metabolites such as organic acids (lactic and acetic acid), amino acids, exopolysaccharides, vitamins, mannitol, bacteriocins, etc., and many of them possess health-promoting effects such as antioxidative, anticancer, reduction in total cholesterol, and anti-inflammatory effects [3,4,5]. The major metabolites produced by LAB through lactic fermentation are either exclusively lactic acid (homo lactic fermentation) or mixture of lactic acid, ethanol, acetic acid, and CO_2_ (hetero lactic fermentation), depending on the species. Other minor compounds such as γ-aminobutyric acid (GABA), ornithine, mannitol, exopolysaccharide (EPS), 2-hydroxyisocaproic acid (HICA) and 3-phenyllactic acid (PLA) are derived from amino acids or sugars by enzyme activities of LAB. The production of some compounds helps LAB hosts to adjust to adverse environments. GABA and mannitol are the examples (see below).

Kimchi fermentation is carried out naturally without starters in most cases, but the use of starters is on the rise recently as the demand for the production of kimchi with consistent quality in an industrial scale becomes important [1]. The quality of kimchi is affected by many factors, including types and ratios of raw materials used and fermentation conditions (temp., pH, salt concentration). Diverse LAB species become dominant at different stages of kimchi fermentation, and the fermentation conditions vary widely. The most important groups of LAB are *Leuconostoc*, *Lactobacillus*, and *Weissella*, and they include *Leuconostoc mesenteroides*, *Leuconostoc citreum*, *Leuconostoc gelidium*, *Lactobacillus brevis* (now, *Levilactobacillus brevis*), *Lactobacillus plantarum* (now, *Lactiplantibacillus plantarum*), *Lactobacillus sakei* (now, *Latilactobacillus sakei*), *Weissella confusa*, *Weissella koreensis,* and *Weissella cibaria* [6,7,8]. Due to the rapid progress in next-generation sequencing and multi-omics technologies including metagenomics, metatranscriptomics, and metabolomics, our understanding on the microbial communities, succession of LAB species, highly expressed genes, and important metabolites at different stages of kimchi fermentation has been improved greatly. All these data are useful not only for understanding the abilities of kimchi LAB to adjust to kimchi fermentation environments but also for finding methods to produce kimchi with improved functionalities. New technologies will also help scientists to find better ways to utilize LAB as probiotics and hosts for producing useful metabolites for industrial applications [9]. Compared to LAB from dairy environments, relatively few studies have been carried out for LAB originated from plant environments including kimchi. Considering the diversity of plant materials used for food fermentation and LAB inhabiting on them, extensive studies are necessary on LAB from plants, which include isolation of novel species from diverse ethnic foods, identification of metabolites, and studies on the biochemical and functional properties of metabolites [9,10]. Since LAB have been used for fermented foods for thousands of years, LAB and their metabolites with GRAS (generally recognized as safe) status have good potential as probiotics and starters. Recently, LAB have received much interest as hosts for producing industrially valuable commodities from renewable feedstocks due to the advantages of LAB and progress in metabolic engineering [11]. In this review, selected metabolites produced by LAB originated from kimchi are discussed for their properties, roles during kimchi fermentation, and their potential and importance for food, feed, and pharmaceutical industries.

## 2. Important Metabolites Produced by Kimchi LAB

### 2.1. Bacteriocins

Bacteriocins are proteins or peptides synthesized at ribosomes as a primary metabolite, which possess antimicrobial activities against closely related species (narrow spectrum) or diverse microorganisms (broad spectrum) [12,13]. Many LAB species secrete bacteriocins, and nisin, produced by some *Lactococcus lactis* strains, is the most well-known and the only commercially produced bacteriocin with approvals from governmental agencies as a food preservative [14]. Although research on bacteriocins has continued for several decades, industrial applications of bacteriocins are still very limited. This is largely due to the lack of stability of bacteriocins for extended periods, higher production costs, and narrow inhibition spectra in repressing spoilage or pathogenic microorganisms when applied to food systems [14]. Despite these shortcomings, bacteriocins, especially from LAB, gain continuous interests due to their potentials as safe food preservatives or alternatives for medically important antibiotics for treating infectious diseases [15]. Bacteriocins from LAB are GRAS compounds and can be incorporated into foods as purified compounds or partially purified forms. Alternatively, culture supernatant can be added, or a producing strain can be used as a starter. Since consumers prefer foods with no or fewer chemical preservatives, bacteriocins will remain as an important topic for food industry. Recent progress in protein engineering will accelerate the efforts to improve bacteriocins into better forms in terms of stability and inhibition spectra. A bacteriocin producing LAB has a good potential as a starter for kimchi fermentation. Selected LAB strains isolated from kimchi and producing bacteriocin are shown in Table 1 (references [16,17,18,19,20,21,22,23,24,25,26,27,28,29,30]). Most bacteriocins in Table 1 are small-sized peptides, 7 kDa or less, except a bacteriocin from *P. pentosaceus* T1, which is 23 kDa in size [30]. Many of them retained full activity after 15 min treatment at 121 °C, indicating their significant heat tolerance and suitability as food preservatives. Considering the small size and significant heat resistance, many bacteriocins produced by kimchi LAB are either class 1 or class 2 bacteriocins [12].

*Leuconostoc mesenteroides* GJ7, a kimchi isolate, produced a bacteriocin whose activity was increased by the presence of sensitive cells [1]. Kimchi fermented with *L. mesenteroides* GJ7 showed desirable properties such as extended shelf-life and improved texture and sensory properties compared to control kimchi (no starter). It is noteworthy that *L. mesenteroides* remained the most dominant species, occupying 70–90% of total viable bacteria during 125 days of storage. The addition of starter to kimchi does not necessarily guarantee that the starter remains the dominant organism throughout the fermentation because the starter organism is often outcompeted by other organisms naturally present in kimchi [31]. In this respect, a bacteriocin producer has an advantage as a starter. Yeasts are regarded undesirable for kimchi fermentation since they are responsible for bad odors and a softened texture observed at later stage of fermentation. Yeasts were not detected in starter kimchi during the entire 125 days, whereas yeasts appeared after 50 days in control kimchi [1]. *Leuconostoc citreum* GR1, another kimchi isolate, produced a bacteriocin, and its bacteriocin activity was increased by the presence of some *Lactobacillus* species [26]. In kimchi fermented with *L. citreum* GR1, *L. citreum* GR1 impressively dominated throughout the initial fermentation period (6 °C, 9–12 days) and the following 2 months of storage at −1 °C. Kimchi fermented with *L. citreum* GR1 showed better sensory properties and higher mannitol content than control kimchi (no starter) and two other kimchi fermented with other LAB strains with antibacterial activities. Similar results were reported by Lee et al. who isolated two *Leuconostoc mesenteroides* strains from kimchi and used them as starters independently [32]. Kimchi fermented with each *L. mesenteroides* strain showed higher mannitol contents than control kimchi (no starter), and the proportions of *Leuconostoc* among total LAB were maintained higher, 70% at 3 weeks [32]. Interestingly, these *L. mesenteroides* strains were selected based on properties such as mannitol production, low acid/gas production, and acid resistance, but bacteriocin production was not considered. Considering the significant dominance (70%) of *L. mesenteroides*, bacteriocin or other antibacterial substances might be produced in these *L. mesenteroides* strains. These results indicate that production of kimchi with extended shelf-life, improved taste, and enhanced functionality such as mannitol production is possible by using a starter, producing a bacteriocin and a metabolite of interest at the same time. Consumption of such kimchi might positively affect the microbiome in intestines since bacteriocins and the producing cells could inhibit sensitive pathogens in the human intestines [33]. An example is Thuricin CD, a narrow spectrum bacteriocin from *Bacillus thuringiensis,* which kills *Clostridium difficile* [34]. The addition of purified Thuricin CD into the fecal microbiota in a colon distal model reduced *C. difficile* effectively without significant impact on the composition of the microbiota. Similar effects can be expected for narrow spectrum bacteriocins from LAB. In addition to antimicrobial activity, bacteriocins may exert other health-promoting effects, such as anti-cancer and immunomodulation. Nisin administered through the oral route showed immunomodulatory effects in a mouse model and human cell line test [15], which showed a possibility that bacteriocins from LAB might exert immunomodulatory activities by affecting production of cytokines [35]. Nisin inhibited the progression of oral squamous cell carcinoma (OSCC), a subset of head and neck squamous cell carcinoma, by interfering the cross-talk between the integrin/FAK and TLR/MyD88 signaling pathway, which accelerates the progression of OSCC [36].

Considering these reports, bacteriocins from kimchi LAB including nisin can be used not only as safe and natural food preservatives but also as therapeutic agents for treating pathogen infections, some types of cancers, or improving general health by modulating immune responses or microbiota in the intestines. Recent progress in protein engineering will accelerate the development of modified bacteriocins with improved properties, and this will lead to new applications of bacteriocins for medical fields. Various nisin derivatives were obtained by replacing amino acids with different amino acids at several positions, and some mutants showed increased inhibitory activities and/or extended inhibition spectra. For example, a nisin A derivative where the 20th amino acid was changed from asparagine into proline (N20P) showed enhanced activity against *Staphylococcus aureus* including a MRSA isolate, and another derivative (M21V) showed higher anti-listerial activity compared to wild-type nisin A [37]. Changes at the 12th lysine or 29th serine residues such as K12A, S29G, or S29A caused the mutants to have broader inhibition spectra against Gram-positive bacteria. Attempts are being made to obtain nisin derivatives with increased stability against proteases or at alkaline pHs [37]. Bacteriocin-producing LAB isolated from kimchi are good candidates as starters for fermented foods and hosts for producing bacteriocins, which can be used for food, feed, and pharmaceutical industries. To accelerate the industrial application of bacteriocins, efforts to isolate LAB producing novel bacteriocins from diverse natural environments including plants and fermented foods should be continued. Data mining of genome sequences effectively complements the traditional screening method for bacteriocin producers. For example, 59 genomes containing putative bacteriocin-encoding gene clusters were identified through data mining of total 382 microbial genomes obtained from a human microbiome project [33]. The actual production of bacteriocins from the organisms should be confirmed with culture supernatant by traditional screening methods such as spot-on-the-lawn or agar well diffusion method.

### 2.2. GABA (γ-Aminobutyric Acid)

GABA is a non-protein amino acid, and a major inhibitory neurotransmitter in the mammalian central nervous system [38]. GABA has been known to exert important physiological functions, including anti-depression, relieving anxiety, anti-hypertension, anti-diabetic effects, renal protection, etc. [39,40]. GABA is commercially available, and in Korea, products containing GABA converted from L-glutamic acid are sold as health functional foods, certified by the Korean government, with blood pressure lowering effects (https://www.foodsafetykorea.go.kr/portal/healthyfoodlife/functionalityView09.do?menu_grp=MENU_NEW01&menu_no=2657, accessed on 8 September 2021). Among microorganisms, LAB are the major GABA producers, and *Lactobacillus brevis* (now, *Levilactobacillus brevis*), often isolated from fermented foods, is the most important species. GABA is produced by irreversible decarboxylation of L-glutamate under anaerobic condition, and glutamate decarboxylase (GAD, EC 4.1.1.15) catalyzes the conversion at an acidic pH (Figure 1). The optimum pH of GAD is between 4.0 and 5.5, and pyridoxal 5-phosphate is required as a cofactor [38,41]. GABA production in LAB is a part of acid stress responses, and cells try to maintain intracellular pH at near neutral by removing internal protons through conversion of L-glutamate into GABA [41,42]. For this reason, many GABA-producing LAB species have been isolated from kimchi. The pH of fresh kimchi, immediately after preparation and not fermented yet, is between 5 and 6, but pH drops rapidly below 5 as fermentation proceeds, and the pH is in the range of 4.2–4.4 at its optimum stage for consumption [43].

GABA content of kimchi fermented naturally, i.e., fermented without added starter, is not high enough to confer the claimed health benefits to consumers. To increase the GABA content of kimchi, use of a starter with a high GABA-producing ability is essential because the addition of chemically synthesized GABA is not preferred by consumers. In addition to the use of a starter, kimchi fermentation should be carried out under carefully controlled conditions where the starter produces the maximum amount of GABA. In this respect, studies are necessary on the isolation of novel LAB strains with high GABA production yields, optimum fermentation conditions for the isolated strains, and methods for providing a precursor of GABA (L-glutamate or monosodium glutamate (MSG)) in sufficient amounts. A selected list of GABA-producing LAB isolated from kimchi or jeotgal is shown in Table 2 (references [44,45,46,47,48,49,50,51,52,53,54,55,56,57,58,59]). Jeotgals are Korean traditional salted and fermented sea foods prepared from small fishes, fish intestines, fish eggs, shrimps, or shellfishes. Jeotgal is often added to kimchi to enhance the flavor and the taste of kimchi [60]. Kimchi fermented with jeotgal showed higher GABA, acetate, lactate and mannitol contents than kimchi without jeotgal [60]. Jeotgals inhabiting LAB are most likely to grow in kimchi and play some roles during kimchi fermentation. It is noteworthy that GABA yields in Table 2 were measured differently among researchers. For example, different growth media for LAB were used, and different concentrations of GABA precursor (MSG or L-glutamate) were used under different conditions. Measurement of GABA content was performed by either the HPLC method or GABase method. Thus, it is difficult to compare GABA yields directly among the LAB strains, and readers are advised to consider this. *Lactobacillus zymae* (now, *Levilactobacillus zymae*) GU240 was isolated from kimchi and the strain produced GABA profusely [59]. Baechu kimchi was prepared by inoculating *Lb. zymae* GU240 as the starter (10^7^ CFU/g kimchi), and L-glutamic acid, MSG, or kelp extract was added separately to kimchi as a precursor of GABA [61]. Kimchi samples were fermented for 20 weeks at −1 °C. Kimchi fermented with *Lb. zymae* GU240 without a precursor did not show an increase in GABA and its GABA content was not different from that of control kimchi (no starter and no precursor) at 20 weeks. Kimchi fermented with the starter and MSG (1%, *w/w*) showed the highest GABA content followed by kimchi fermented with starter and kelp extract (3%, *w/w*). L-glutamic acid was the least effective precursor, and its addition (1%, *w/w*) inhibited the growth of starter due to its strong acidity. Although GABA content of kimchi fermented with the starter and MSG increased compared to that of control kimchi, the increase was not great, 194% of control kimchi. GABA content of kimchi (starter + MSG) was at the highest (120.3 mg/100 g kimchi) at 8 weeks, and that of control kimchi was also at the highest (62.0 mg/100 g kimchi) at 8 weeks. Other kimchi samples showed the highest GABA contents at 10 weeks, and those of kimchi (starter + kelp extract), kimchi (starter + L-glutamate), and kimchi (starter only) were 82.2 mg/100 g kimchi, 76.6 mg/100 g kimchi, and 69.3 mg/100 g kimchi, respectively. Kimchi fermented with the starter and MSG achieved the highest scores in sensory evaluation. In another study, a GABA producing *Lactobacillus buchneri* (now, *Lentilactobacillus buchneri*) was inoculated to baechu kimchi (10^5^–10^7^ CFU/g), and mukeunjee kimchi (overacidified kimchi) was prepared by fermentation for 3 days at 30 °C [50]. GABA content was measured after fermentation and compared with that of control kimchi (no starter). The GABA content of mukeunjee kimchi fermented with the starter was 61.7 ± 9.0 mg/100 g, whereas that of mukeunjee kimchi without the starter (control) was 8.1 ± 0.5 mg/100 g [50]. In this work, a precursor was not added, and this seemed to be the reason for the low yield of GABA.

More research is necessary to increase GABA content of kimchi to meaningful levels, and these should include methods for providing a GABA precursor economically and without adverse effects on the sensory properties of kimchi, and selection of GABA producers which grow rapidly and become dominant over other LAB species naturally present among raw materials of kimchi. Additionally, a standardized procedure for measuring the GABA content of kimchi needs to be established, including collection time and collection method of kimchi samples, and the GABA assay method. One question which remains to be answered is how to alleviate the dependence of GAD on pyridoxal 5-phosphate for the activity. The addition of pyridoxal 5-phospate is not practical considering its cost. The isolation of LAB strains whose GADs do not require pyridoxal 5-phosphate or the construction of mutant GADs which no longer require pyridoxal 5-phosphate might be the answer. Detailed understanding on the GAD structure and the roles of amino acids involved in binding with a substrate or cofactor is critical for the construction of such mutants and mutants with increased GAD activities or improved stabilities [41].

In many GABA-producing LAB isolated from kimchi or jeotgal, *gadB*-encoding GAD forms an operon with *gadC*-encoding L-glutamate/GABA antiporter which is responsible for the uptake of L-glutamate into the cytoplasm and the simultaneous exclusion of GABA. The gene order (*gadCB*) was confirmed experimentally for some isolates [44,48,54,58,59]. Properties of purified GADs from LAB isolated from kimchi or jeotgal are shown in Table 3 (references [44,45,46,47,48,49,50,51,52,53,54,55,56,57,58,59,60,61,62]). Unlike *gadB*, little research has been performed with *gadC. gadR* encodes a positive transcription regulator for GABA production and is usually located upstream of the *gadCB* operon. *Lb. brevis* (now, *Levilactobacillus brevis*) D17 was isolated from Chinese acidic liquor product, and the strain showed enhanced *gadR* expression and also enhanced acid resistance [63]. *Lb. brevis* D17 produced much more GABA than *Lb. brevis* ATCC 367 when both strains were grown under the pH-controlled and mixed-feed conditions. A *gadR* deletion mutant was obtained from *Lb. brevis* ATCC 367 and *gadB* and *gadC* were not expressed in the mutant in the presence of MSG. However, both genes together with *gadR* were expressed at higher levels in *Lb. brevis* ATCC 367 under the same conditions [63]. The results confirmed that GadR plays an important role in GABA production and show a possibility of increasing GABA production by modulating *gadR* expression. Further research is necessary on the detailed roles of *gadC* and *gadR* for GABA production, which will help to better understand GABA production in LAB.

### 2.3. Ornithine

Ornithine is a non-proteinogenic amino acid and a central intermediate in the urea cycle. Ornithine is derived from arginine, an essential amino acid for man and animals. Ornithine has several important physiological functions such as anti-obesity, stimulation of growth hormone, promoting muscle growth, anti-fatigue, treating cirrhosis, etc. and is currently available as a neutraceutical supplement, popular with athletes [64]. It was suggested that lactobacilli in the gut might contribute to the homeostasis of the gut mucus layer by producing L-ornithine from L-arginine, which in turn stimulates tryptophane metabolism in gut epithelial cells, and an increase in gut immune cells [65]. Ornithine is derived from arginine by arginine deiminase (ADI) pathway enzymes in LAB [64]. The conversion of arginine to ornithine is a part of acid stress responses in addition to serving as a source of energy, carbon, and/or nitrogen for some LAB, such as *Levilactobacillus*
*brevis* [66]. Arginine enters into the cell by arginine/ornithine antiporter (ArcD, *arcD* gene product) with the concomitant excretion of ornithine from the cytoplasm (Figure 2). Inside the cell, arginine is converted into citrulline and ammonia by arginine deiminase (ArcA, *arcA* gene product). Citrulline is converted into ornithine and carbamoyl phosphate in the presence of a phosphate by ornithine transcarbamylase (OTC, *arcB* gene product). Carbamoyl phosphate and ADP are converted into ammonia, carbon dioxide, and ATP by carbamate kinase (CK, *arcC* gene product), and ATP is generated during this step, which helps the host cell to grow under anaerobic and acidic conditions [66]. Not all LAB carry all these *arc* genes, and some LAB carry just 1 or 2 genes. For example, *Lactobacillus curvatus* (now, *Latilactobacillus curvatus*) carries *arcD* only and *Leuconostoc kimchi* carries just *arcB* [64]. This explains the observation that the addition of arginine into culture encouraged the growth of some species but not others. A different type of arginine/ornithine antiporter, ArcE (*arcE* gene product), was found in *Streptococcus pneumoniae*, and ArcE bound to citrulline with higher affinity than with arginine, indicating the operation of citrulline catabolism in this host [67]. These findings indicated the importance of L-arginine/L-citruilline for the growth of some LAB species such as *L. brevis* under acidic environments. In such LAB species, the roles of ornithine synthesis are the same with those of GABA synthesis from L-glutamate. Ornithine or GABA helps host cells to maintain the intracellular pH as less acidic by ammonia production or removal of intracellular protons. Acquisition of extra ATP is another advantage for hosts [68].

Red pepper powder (RPP) is an important ingredient for kimchi, and RPP has been known to stimulate the growth of *Weissella* species over other LAB species during kimchi fermentation, although RPP itself does not contain *Weissella* species [69,70]. The addition of RPP encourages the growth of *Weissella* species, and *Weissella* species possess strong arginine deiminase activities, resulting in enhanced ornithine levels in kimchi [70]. This finding is useful for the production of kimchi enriched with ornithine. A strain with the complete ADI pathway enzymes is ideal as the starter, which includes *L. brevis* or *Weissella* species such as *W. cibaria*, *W. confusa*, or *W. koreensis*. Selected ornithine producing LAB isolated from kimchi are shown in Table 4 (references [71,72,73,74]). Kimchi fermented with *W. koreensis* OK1-6 showed anti-obesity effects in a mouse model where high-fat diet induced obese mice were used [75]. A group fed with a high-fat diet containing 3% kimchi fermented with the starter showed significantly decreased lipid, insulin, and leptin concentrations together with a reduced expression of genes involved in lipid anabolism compared with control groups (high-fat diet group and high-fat diet with 3% kimchi without starter group). In another study, mice fed with a high-fat diet and *Lactobacillus brevis* (now, *Levilactobacillus brevis*) OPK-3 (10^9^ CFU/day), a kimchi isolate showing arginine deiminase activity, showed lower body weight, lower epididymal fat tissue mass, and reduced gene expressions of pro-inflammatory cytokines compared to the control (high-fat diet group) [71]. The results suggested that *L. brevis* OPK-3 inhibited the induction of inflammation in addition to inhibition of weight gain. Mun et al. produced a functional rice bran product enriched with ornithine and citrulline by inoculating *W. koreensis* DB1, an isolate from kimchi, into rice bran slurry (20%, *w/v*, pH 6.0) fortified with arginine (3%, *w/v*), glucose (2%, *w/v*), and corn steep liquor (3%, *w/v*) [72]. After 48 h fermentation at 30 °C, the ornithine and citrulline contents were 43,074.13 mg/kg and 27,336.37 mg/kg, respectively, significantly higher than those of previous reports, and high enough to be used directly as a healthcare product without further concentration. This is a good example of converting a by-product from the food industry such as rice bran, a waste derived from rice polishing process with low commercial value, into a value-added product with an anti-obesity function. Although authors did not try to produce kimchi enriched with ornithine via the use of *W. koreensis* DB1, it seems possible to produce kimchi or other fermented foods with high ornithine contents by the same approach.

### 2.4. Mannitol

Mannitol is a six-carbon sugar alcohol, and present among plants and microorganisms. In bacteria, mannitol is a compatible solute, protecting host cells against various stresses including hyperosmotic stress [76]. LAB, both homo- and heterolactic fermenters, produce mannitol but heterolactic fermenters such as *Leuconostoc* species produce more mannitol [77]. In heterolactic fermenters, fructose is reduced to mannitol by mannitol dehydrogenase (EC 1.1.1.67). Fructose serves not only as an electron acceptor but also as a carbon source for growth. Fructose is phosphorylated into fructose 6-phosphate (F-6-P) by fructokinase, and F-6-P is converted into glucose 6-phosphate (G-6-P). G-6-P is metabolized via the pentose phosphate pathway, converted into 1 glyceraldehyde 3-phosphate and 1 acetyl phosphate (Figure 3) [76]. A lactate is produced from glyceraldehyde 3-phosphate via the Embden–Meyerhof pathway, and either acetate or ethanol is produced from acetyl phosphate. The reduction of fructose into mannitol generates NAD^+^ from NADH, which reduces the burden of regenerating NAD^+^ and drives the conversion of acetyl phosphate into acetate rather than ethanol with concurrent production of 1 ATP [78]. Mannitol production yield was increased by cultivating *Leuconostoc* species in medium containing both glucose and fructose [76], or by the use of a mutant where fructose utilization as an energy source was blocked. Helanto et al. constructed a *Leuconostoc pseudomesenteroides* mutant by chemical treatment and the mutant showed 10% of wild-type fructokinase activity. The mutant produced more mannitol compared to wild type (85% versus 74%, mol/mol) in the presence of glucose and fructose because F-6-P formation was prevented, and more fructose was converted into mannitol [79]. This is an example of obtaining strains producing more mannitol by modifying the metabolic pathway. Homolactic fermenters such as *Lactococcus lactis* produce mannitol from fructose 6-phosphate, which is reduced into mannitol 1-phosphate by mannitol 1-dehydrogenase (EC 1.1.1.17). Then mannitol 1-phosphate is dephosphorylated into mannitol by mannitol phosphatase [76]. LDH (L-lactate dehydrogenase)-deficient *L. lactis* strains were used for mannitol production and NAD^+^ regeneration was performed by mannitol 1-dehydrogenase. The highest mannitol production was observed in an LDH-deficient *L. lactis* strain where a *mtlD* encoding mannitol 1-dehydrogenase from *Lactobacillus plantarum* and a gene encoding mannitol phosphatase from *Eimeria tenella* were introduced and overexpressed by the NICE (nisin-controlled gene expression) system [80].

Mannitol confers a cool, sweet taste to foods, and the sweetness is about half due to sucrose. Mannitol is not cariogenic, and does not increase blood sugar levels significantly, showing a low glycemic index, and thus being useful for diabetics [81]. Mannitol has a low caloric value (1.6 kcal/g) compared with sucrose (4 kcal/g) and is thus used as a sweetener for diet foods and chewing gum [82]. Mannitol is used in medicine as an osmotic diuretic, relieving symptoms and pains caused by edema, hydrocephalus, and glaucoma [83]. Currently, mannitol is produced by three different ways: chemical synthesis, enzymatic conversion, and microbial fermentation. Although mannitol is mostly produced by chemical methods, microbial fermentation has advantages for large-scale production because it does not require complex separation steps to remove by-products such as sorbitol, and does not require highly purified substrate, and expensive cofactor NADH [81,84]. *Leuconostoc* species are heterolactic fermenters, and the most important group during the early and middle stages of kimchi fermentation [8]. Therefore, *Leuconostoc* species such as *L. mesenteroides* isolated from kimchi seem to be ideal hosts for mannitol production by fermentation. Such strains are also useful as kimchi starters since mannitol confers a refreshing taste to kimchi, and kimchi starters producing mannitol could be valuable probiotics considering the beneficial effects of mannitol. Otgonbayar et al. compared nine *Leuconostoc* strains previously isolated from kimchi for mannitol production and found that *L. citreum* KACC 91348P grew quickly in MRS broth and produced more mannitol than others [85]. The highest mannitol production (14.83 g/L/h) was observed when *L. citreum* KACC 91348P was cultivated in a 2 L batch fermenter at 30 °C and a constant pH of 6.5. Jung et al. prepared baechu kimchi using a mannitol-producing *L. mesenteroides* strain as the starter (10^7^ cells/g kimchi), and the mannitol content of starter kimchi was higher than that of non-starter kimchi during 40 days of fermentation at 4 °C [86]. During fermentation, mannitol content of kimchi increased as the same pattern with total bacterial cells, whereas fructose content decreased. Kim et al. prepared baechu kimchi by inoculating mixed starters consisting of *Lactococcus lactis* and *Leuconostoc citreum* [87]. *L. lactis* WiKim0098 showed strong antimicrobial activity and *L. citreum* WiKim0096 produced more mannitol than other LAB tested. When kimchi was fermented with mixed starters, overacidification of kimchi was delayed because of growth inhibition of LAB naturally present in kimchi ingredients by *L. lactis* WiKim0098 compared to control kimchi. The mannitol content of starter kimchi was 5.1 mg/mL, higher than that of control kimchi (3.4 mg/mL) [87]. So far, kimchi fermentation has not been reported with intentionally added fructose. If fructose is added to kimchi, more mannitol is likely to be produced, and it will be interesting to examine the properties of kimchi, such as mannitol content, growth of LAB, and sensory properties. Rice et al. reported an interesting application of a *Leuconostoc* strain for production of a healthy beverage [78]. A *Leuconostoc citreum* was isolated from sour doughs and inoculated into apple juice as a starter. Fructose and sucrose contents of apple juice were decreased significantly (83% reduction) with the concomitant increase in mannitol (61.6 g/L) after 48 h fermentation in a 1 L bioreactor system. The product contained less sugars, but still maintained sweetness due to mannitol, showing a possibility of producing healthier beverages by converting fructose into mannitol.

### 2.5. Exopolysaccharides

Exopolysaccharides (EPSs) are biopolymers synthesized by microorganisms such as bacteria, fungi, and blue-green algae [88]. EPSs are located outside of cells, either attached to the cell surface tightly or loosely, or detached from the cell. EPSs have several protective roles for hosts, including protection from viruses, predators, dehydration, or toxic chemicals [89]. EPSs from LAB have been the subject of many studies because they have important effects on the rheological, organoleptic, and functional properties of foods. Currently, EPSs from LAB, considered GRAS compounds, have been used in food, pharmaceutical, cosmetic, and chemical industries. Recently, EPSs and their modified derivatives are receiving attention due to their potential as carriers for drugs [90]. EPSs are classified into two types: homopolymers (hoPO) and heteropolymers (htPO). The former consists of a single monosaccharide such as glucose or fructose, and the latter consists of at least two different monosaccharides connected at different positions [91]. The sizes and properties of EPSs are variable depending upon the host bacteria, substrates, and growth environments. Dextran is the most well-known hoPO produced by some *Leuconostoc, Lactobacillus*, and *Streptococcus* species grown on sucrose-containing medium. Dextran consists of α-D-glucose units connected via α-1→6 glycosidic bonds with a few branches connected via α-1→3 bonds, and occasionally α-1→2 or α-1→4 linkages [88,92]. In the food industry, dextran is used for increasing the viscosity and volume of foods, moisture retention, prevention of crystallization, and improving the texture of foods [93]. In medicine, dextran is used as an antithrombotic, to reduce blood viscosity, and as a volume expander in hypovolaemia. Dextran also receives interest as a potential drug carrier due to its excellent water solubility, biocompatibility, biodegradability, and low toxicity. Dextran can be modified to various forms, such as nanoparticles, gel, microspheres, sponge, liposome, etc., to carry a drug and release it slowly in body [90]. EPSs from LAB show anti-cancer, anti-inflammatory, anti-oxidant, anti-viral, anti-biofilm, anti-diabetic, and immunomodulatory activities [94]. EPSs from LAB show anti-cancer activities via various mechanisms including anti-proliferation, apoptosis induction, cell cycle arrest, mutagenicity inhibition, oxidative stress modulation, angiogenesis inhibition and inflammatory amelioration [95]. The anti-cancer effects of EPSs vary depending on the production host, chemical structures, and purity of EPS, and further research is necessary for elucidation of the exact mechanisms. *Lactobacillus plantarum* (now, *Lactiplantibacillus plantarum*) LRCC5310 was isolated from kimchi, and its EPS showed an antiviral effect against the human rotavirus Wa strain in vitro [96]. Oral administration of the EPS into mice reduced the duration of diarrhea, viral shedding, and destruction of enteric epithelium integrity in infected mice [96]. *L. mesenteroides* BioE-LMD18, an isolate from kimchi, produced an EPS of 123.8 kDa in size, which showed immunomodulatory effects in RAW 264.7 murine macrophage cells previously treated with lipopolysaccharide (LPS) from *Escherichia coli* [97]. EPS treatment reduced the amount of IL-6, a pro-inflammatory cytokine, by 32.1%, but increased the amount of IL-10, an anti-inflammatory cytokine, by a dose-dependent manner in LPS-treated cells. The EPS from *L. mesenteroides* BioE-LMD18 was a htPO, consisting of mannose (8.71%), arabinose (0.07%), galactose (1.22%), fucose (10.21%), and glucose (79.8%). Similar immunomodulatory effects were previously reported for dextrans from four *L. mesenteroides* and one *Lactobacillus sakei* isolates [98]. When these dextrans were independently given to human monocytic leukemia cell line THP-1, the amount of pro-inflammatory TNF-α was reduced, whereas that of IL-10 was increased. More studies are necessary on the relationships between the structures and the biofunctionalities of EPSs from LAB. However, data so far are enough to claim that EPSs such as dextran have great potential for food and pharmaceutical industries.

LAB-producing EPSs with desirable properties are promising probiotics and also starters for fermented foods such as kimchi and fermented dairy products. For example, two *Weissella cibaria* strains were isolated from kimchi, and they showed desirable properties including resistance against acid and bile salts, sensitivity to antibiotics, strong adhesion to intestinal epithelial cells, inhibition of some food pathogens, high butyric acid production, and EPS production [99]. Such strains have potentials as starters for functional foods and/or probiotics, but the real efficacy of each strain should be confirmed experimentally. EPSs produced by LAB are the products of complex metabolic processes involving many genes responsible for the syntheses and modifications of sugars and their regulations. In addition to genetic elements, the availability of sugars and environments also affect the types of EPSs produced. Scientists have been trying to overproduce EPSs or produce EPSs with improved properties through metabolic engineering such as insertion of heterologous genes and/or inactivation of inherent genes, thus modifying biosynthetic pathways or creating new pathways for the optimum synthesis of an EPS [100,101,102]. For example, a gene encoding NADH oxidase from *Streptococcus mutans* was introduced into *Lactobacillus casei* LC2W, an EPS producer. The recombinant strain produced 46% more EPS compared to wild type [101]. The increase was due to less lactate production because the heterologous NADH oxidase oxidized NADH into NAD^+^, thus reducing the need of pyruvate reduction into lactate [101]. This kind of so-called “cofactor engineering” is a common strategy used for the genetic engineering of LAB strains, and the purpose is the production of an alternative product such as L-alanine, diacetyl, acetoin, mannitol, EPS, or 2,3-butanediol instead of lactate. The same strategy is useful for the overproduction of EPSs by strains isolated from kimchi. It can be envisioned that a LAB strain producing a functional EPS, a bacteriocin, GABA, mannitol, etc. simultaneously can be constructed by use of state-of-the art technologies and used for the production of foods with multi-functionalities. Alternatively, a group of strains will be constructed, and each strain is tailored for the production of a specific metabolite and used as necessary. Advances in metabolic engineering combined with multi-omics technologies are likely to make this scenario come true in the near future [102].

### 2.6. 2-Hydroxyisocaproic Acid and 3-Phenyllactic Acid

2-Hydroxyisocaproic acid (HICA) and 3-phenyllactic acid (PLA) are metabolites produced by bacteria including LAB, and they are receiving attention due to their antimicrobial activities and other useful properties. Both are end products from amino acids catabolism31. HICA is derived from L-leucine and PLA from L-phenylalanine [103,104]. For HICA production, leucine is first converted into 2-ketoisocaproic acid (KICA) by a branched-chain amino acid aminotransferase which transfers the amino group of leucine to α-ketoglutarate. Then KICA is reduced into HICA by hydroxyisocaproate dehydrogenase (HicD, product of *hicD*) (Figure 4A) [105]. For PLA production, phenylalanine is converted into phenylpyruvic acid (PPA) by an aminotransferase, and then PPA is reduced to PLA by a dehydrogenase such as lactate dehydrogenase (Figure 4B) [106]. NADH provides the reducing power for the reaction. Recently, it was proposed that PLA might be produced not only by the catabolism of phenylalanine but also by the anabolic pathway for phenylalanine synthesis [104]. Both HICA and PLA possess antibacterial and antifungal activities, and they are considered as food preservatives or topical agents for treating bacterial and fungal infections. When beef cuts were surface sprayed with PLA (1.5%), all inoculated *E. coli* O157:H7 and *Salmonella* Typhimurim cells were inactivated [107]. HICA is also known to increase muscle mass and help recovery after exercise, and is used as a supplement for athletes [108]. However, a contradictory report was published where administration of HICA to young adults showed no effect on muscle growth [109]. PLA has a potential to serve as a building block for poly (PLA), which has some desirable properties as a degradable bioplastic [110]. 3T3-L1 preadipocytes treated with PLA showed increases in glucose uptake and adipocyte differentiation by activation of PPAR-γ2 [111]. From these observations, it was suggested that PLA might have a potential as a drug for preventing type-2 diabetes mellitus. Immunomodulatory effects of PLA were also suggested. The consumption of fermented foods such as sauerkraut, which contained PLA in μM concentration, caused the increase in the blood PLA level, which in turn affected monocyte functions and caused immunomodulation via the binding of PLA to hydroxycarboxylic acid receptor 3 at monocytes and immune cells of man [112].

The amount of HICA and PLA produced are different among LAB species. When LAB species isolated from kimchi were compared for HICA production after cultivation in MRS broth, *Lb. plantarum* produced the largest amount of HICA followed by *L. mesenteroides,* but *Lb. sakei* and *Pediococcus pentosaceus* produced small amounts [105]. The amount of HICA produced by a specific LAB species seemed to be closely related to the number of *hicD* gene and its expression level. It was shown that *Lb. plantarum* had three or four *hidD* genes and *L. mesenteroides* had one gene on the genomes. However, no *hicD* gene was located on the genomes of *Lb. sakei* and *P. pentosaceus* [113]. Putative *hicD* genes are located on the genomes of many LAB species, but their functions have not been confirmed experimentally in most cases. In *Lb. plantarum* and *L. mesenteroides, hicD* expressions were increased at acidic conditions, and higher expressions were observed at pH 4.5 compared to at pH 5.5 [105]. Since overall functionality of a fermented food is determined by the sum of all effective metabolites [11], the use of multiple starters, where each member provides a specific function, may be an effective method to improve the overall functionalities of fermented foods as long as starters are compatible with each other.

HICA effectively inhibited the growth of *Enterococcus faecalis,* a causative organism of dental cary [114]. HICA inhibited the growth of planktonic *Candida albicans* cells and also biofilm formation. The reduction in pre-grown biofilm was most effective at an acidic pH, and the mutagenic acetaldehyde was not produced from glucose at acidic pH [115]. HICA extended the shelf-life of bread by inhibiting the growth of fungi in sourdough [103]. HICA and PLA clearly inhibited the growth of mold in bread, but the amount of total carboxylic acids, including HICA and PLA of a specific strain, did not match exactly with the strength of inhibition by that strain, indicating involvement of other unknown metabolites for the inhibition [103]. HICA acted as a fungicide and killed *Candida* and *Aspergillus* species at high concentration (72 mg/mL) and acted as a fungistatic at lower concentrations [116]. HICA and PLA are produced during kimchi fermentation by LAB. However, HIKA contents of five commercial kimchi stored for 2–4 weeks were low, ranging from 7.1 ± 0.1 to 22.6 ± 0.4 μg/mL [105]. Antimicrobial activity is hardly expected at this low level of HICA. However, LAB produce other inhibitory metabolites, such as organic acids, bacteriocins, hydrogen peroxide, and small-sized inhibitory compounds such as reuterin. All these compounds work synergistically to inhibit pathogens and spoilage microorganisms. Thus, the ability of a strain to produce HICA and/or PLA is an advantage because it is likely to serve as another hurdle for pathogens. Since the production of HICA and PLA is affected by the activities of responsible enzymes such as HisD and the availability of corresponding amino acids (leucine and phenylalanine), it seems possible to increase the content of HICA and PLA of kimchi or other fermented foods to certain levels by use of starters with high enzyme activities and with precursor amino acids in sufficient amounts. The content of HICA in kimchi is affected by the dominant LAB species. The addition of *L. mesenteroides* or *Lb. plantarum* (10^7^ CFU/g kimchi) to kimchi increased HICA content compared to control kimchi [105]. For this reason, it was suggested that HICA content of kimchi might be used as an indicator for kimchi fermentation [105]. Similarly, the amount of PLA in kimchi or other fermented foods is likely to be closely related to the dominant LAB species.

## 3. Conclusions

Plants inhabiting LAB have great potential not only as starters for fermented foods such as kimchi, sauerkraut, and pickles but also as probiotics for humans, animals, poultries, and fishes. LAB also have good potential and receive interest as cell factories for the production of value-added products, including vitamins, 1, 3-propandiol, diacetyl, hyaluronic acid, exopolysaccharides, platform chemicals for bioplastics and biofuels, antibodies, etc. [11] LAB transform nutrients in foods into more bioactive forms or more easily absorbed forms in human intestines by enzyme activities during fermentation. Biotransformation (bioconversion) capabilities of LAB are useful for the precise production of valuable chemicals from their precursors [117]. Considering these advantages, the importance of LAB, especially from plant sources which have been relatively less utilized, will increase in food and other related industries. Although important LAB species involved in kimchi fermentation have been identified, and many of their genomes have been sequenced, detailed knowledge on the functions and potentials of metabolites are still scarce, especially for their effects inside human bodies.

Kimchi serves as a model food for studying the biochemical, physiological, and genetic strategies of LAB species to adapt to acidic, low-temperature, and anaerobic kimchi environments. Metabolites such as GABA, ornithine, bacteriocins, exopolysaccharides, etc. are the products resulting from efforts of LAB to thrive and outcompete other microorganisms under kimchi environments. Therefore, it is necessary to have a deep understanding of the metabolites including the production pathways, key enzymes, regulatory mechanisms, and the roles and potentials as functional materials if a better or novel utilization of LAB is tried. Isolation of novel LAB strains with some functionalities is important for the development of products with commercial value. For example, one popular health functional product certified by the Korean government is based on *Lb. plantarum* CJLP133, which was obtained through screening of 3500 LAB isolates from kimchi. The administration of the strain improves conditions of sensitive skin such as atopic dermatitis, and the effect was proven scientifically [118]. Progress in bioinformatics, such as the identification of potentially useful genes through data mining of genome sequences, metabolic engineering, and genome editing techniques such as CRISPR/Cas9, will make it possible to construct novel strains designed for the optimum production of specific metabolites via food-grade manners; thus, constructed strains can be used directly for food fermentation as starters or probiotics. Similarly, strains producing bacteriocins inhibiting pathogens or eliciting immune responses in human intestines are likely to be constructed and used to improve human health. The recent surge of interest in the human microbiome will accelerate research on probiotics and the expected results contribute to finding novel methods to improve our health.

## Figures and Tables

**Figure 1 foods-10-02148-f001:**
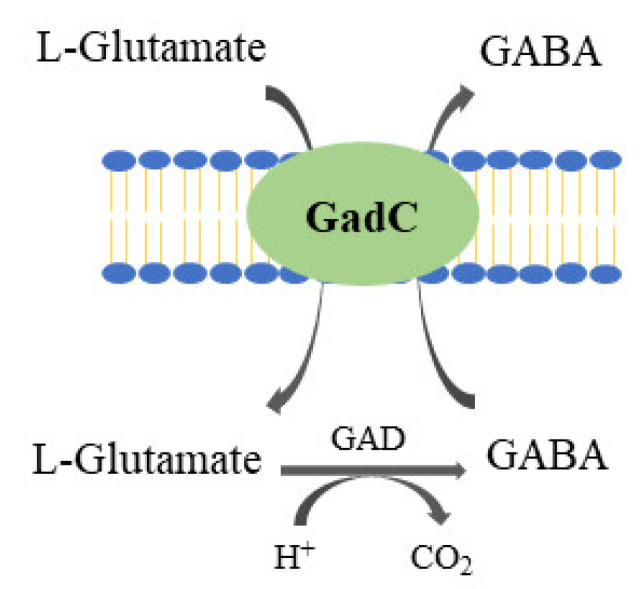
Schematic representation of GABA production via decarboxylation of L-glutamate catalyzed by glutamate decarboxylase. GAD, glutamate decarboxylase; GadC, glutamate/GABA antiporter.

**Figure 2 foods-10-02148-f002:**
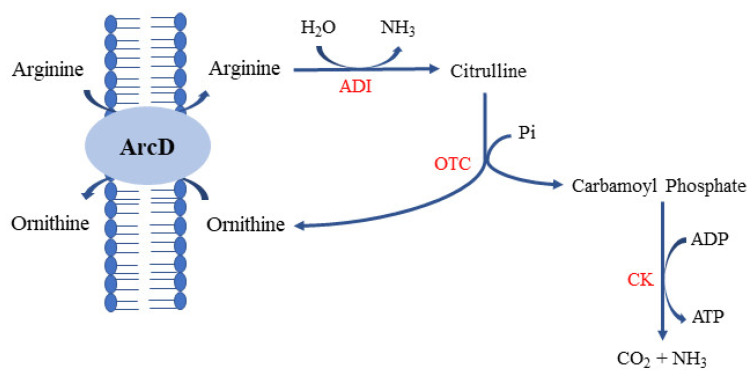
Schematic representation of the arginine deiminase (ADI) pathway. ADI, arginine deiminase encoded by *arc*A; OTC, ornithine transcarbamylase encoded by *arc*B; CK, carbamate kinase encoded by *arc*C; ArcD, arginine/ornithine antiporter encoded by *arc*D.

**Figure 3 foods-10-02148-f003:**
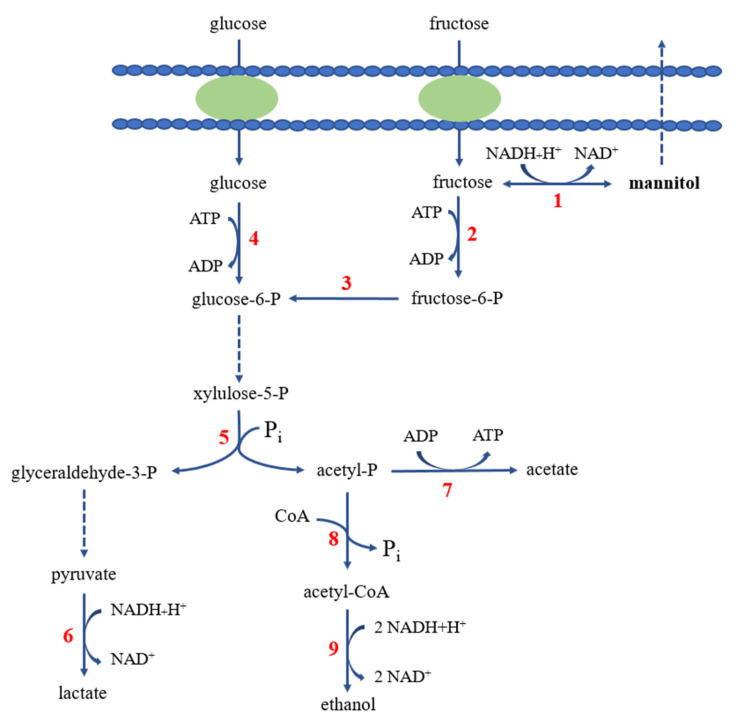
Mannitol production from fructose in heterolactic fermenters such as *Leuconostoc* species. Pathway for fructose utilization is also shown. 1, mannitol dehydrogenase; 2, fructokinase; 3, glucose phosphate isomerase; 4, glucokinase; 5, phosphoketolase; 6, lactate dehydrogenase; 7, acetate kinase; 8, phosphate acetyltransferase; 9, acetaldehyde dehydrogenase and alcohol dehydrogenase [76]. Figure 3 was modified from Figure 2 of reference [76] with the permission from Elsevier at Sep. 9, 2021 (5144670406141).

**Figure 4 foods-10-02148-f004:**
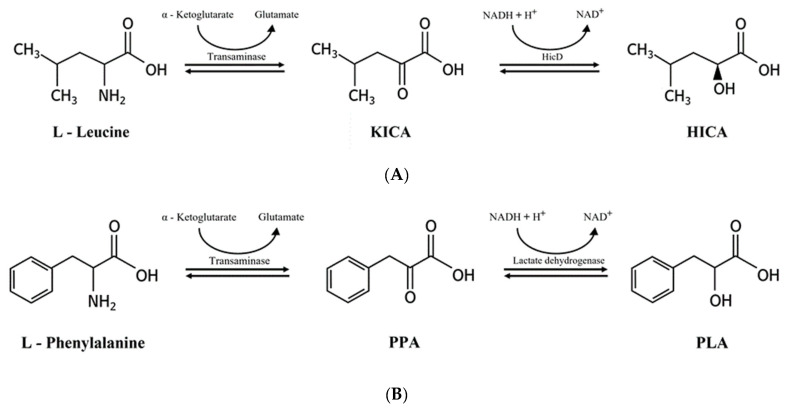
Production of HICA (**A**) and PLA (**B**) from corresponding amino acid precursor, L-leucine and L-phenylalanine, respectively. KICA, 2-ketoisocaproic acid; HICA, 2-hydroxyisocaproic acid; PPA, phenylpyruvic acid; PLA, 3-phenyllactic acid.

**Table 1 foods-10-02148-t001:** Properties of selected bacteriocins produced by LAB isolated from kimchi.

Strain	pH Stability	Temperature (°C) Stability	Molecular Size (kDa)	Reference
*Lactiplantibacillus paraplantarum* C7	2 to 8 ^a^, 9 ^b^, 10 ^c^	100 °C, 20 min, 121 °C, 10 min ^b^, 100 °C, 30 min, 121 °C, 30 min ^c^	3.8	[16]
*Lactiplantibacillus plantarum* J9	3 to 10 ^a^	100 °C, 60 min, 121 °C, 15 min ^a^	less than 6.5	[17]
*Lactococcus lactis* BH5	2 to 9 ^a^	90 °C, 30 min ^a^, 100 °C, 30 min ^d^	3.7	[18]
*Lactococcus lactis* ET45	3 to 5 ^a^, 7 to 11 ^c^	121 °C, 60 min ^a^	4.5	[19]
*Lactococcus lactis* KU24	3 to 7 ^a^, 8 to 9 ^c^	100 °C, 30 min ^a^, 121 °C, 15 min ^c^	6.5	[20]
*Lactococcus lactis* LAB3113	2 to 10 ^a^	100 °C, 30 min, 121 °C, 20 min ^d^	10.5	[21]
*Lactococcus lactis* subsp. *lactis* H-559	2 to 11 ^e^	100 °C, 10 min ^a^, 100 °C, 30 min, 121 °C, 10 min ^c^, 121 °C, 20 min ^d^	3.3	[22]
*Lactococcus lactis* subsp. *lactis* J105	3 ^a^, 2, 4 ^b^, 5 ^c^, 6 to 9 ^d^	4 °C, 24 h, 100 °C, 1 h, 110 °C, 10 min, 121 °C, 15 min ^e^	3.4	[23]
*Latilactobacillus curvatus* SE1	2 to 11 ^e^	100 °C, 60 min ^e^	14	[24]
*Latilactobacillus sakei* B16	2 to 9 ^e^	100 °C, 30 min, 121 °C, 15 min ^e^	ND *	[25]
*Leuconostoc citreum* C2	3 to 4 ^a^, 5 to 7 ^d^	50 °C, 24 h ^a^, 70 °C, 24 h ^d^	ND	[26]
*Leuconostoc citreum* GJ7	2.5 to 9.5 ^a^	70 °C, 24 h, 100 °C, 30 min, 121 °C, 15 min ^a^	3.5	[27]
*Leuconostoc citreum* GR1	3 to 4 ^a^, 5 to 7 ^d^	70 °C, 24 h, 100 °C, 30 min, 121 °C, 15 min ^a^	ND	[26]
*Leuconostoc lactis* SD501	2 to 10 ^e^	121 °C, 15 min ^e^	7	[28]
*Pediococcus pentosaceus* K23-2	2 to 7 ^a^, 8 ^c^	95 °C, 30 min, 121 °C, 15 min ^a^	5	[29]
*Pediococcus pentosaceus* T1	4 to 8 ^e^	110 °C, 20 min ^e^	23	[30]

^a^ 100% activity remained after treatment. ^b^ 75% or more activity. ^c^ 50% or more activity. ^d^ less than 50% activity. ^e^ activity remained but not shown numerically. * ND, not determined.

**Table 2 foods-10-02148-t002:** GABA production yields by LAB isolated from kimchi or jeotgal.

Strain (Microorganisms)	GABA Content	MSG Concentration	Reference
*Enterococcus avium* M5 ^a^	18.47 mg/mL	MRS + 3% MSG	[44]
*Enterococcus faecium* JK29	14.86 mM	MRS + 0.5% MSG	[45]
*Lactiplantibacillus plantarum* K154	201.78 µg/mL	MRS + 3% MSG	[46]
*Lactococcus lactis* subsp. *lactis* B	3.68 g/L	MRS + 1% MSG	[47]
*Latilactobacillus sakei* A156 ^a^	15.81 mg/mL	MRS + 3% MSG	[48]
*Latilactobacillus sakei* OPK 2-59	58.88 mM	MRS + 1% MSG	[49]
*Lentilactobacillus buchneri*	5.83 mg /mL	MRS + 50 mM glutamate	[50]
*Lentilactobacillus buchneri* MS	251 mM	MRS + 5% MSG	[51]
*Levilactobacillus brevis* 340G	68.77 mM	MRS + 3% MSG	[52]
*Levilactobacillus brevis* 877G	18.51 mmol/L	MRS + 1% MSG	[53]
*Levilactobacillus brevis* G144 ^a^	14.58 mM	MRS + 3% MSG	[54]
*Levilactobacillus brevis* HYE1	18.76 mM	MRS + 2.38% MSG	[55]
*Levilactobacillus brevis* K203	44.4 g/L	MRS + 6% L-glutamate	[56]
*Levilactobacillus brevis* NPS-QW-145	25.83 g/L	MRS + 7% MSG	[57]
*Levilactobacillus brevis* NPS-QW-267	24.99 g/L	MRS + 7% MSG	[57]
*Levilactobacillus brevis* T118 ^a^	ND ^b^	MRS + 3% MSG	[58]
*Levilactobacillus zymae* GU240	16.94 mg/mL	MRS + 3% MSG	[48,59]

^a^ strain isolated from jeotgal. ^b^ ND, not determined.

**Table 3 foods-10-02148-t003:** Properties of purified GADs from LAB isolated from kimchi or jeotgal.

Strain	MSG Concentration	Operon Structure	GAD Size (aa)	Optimal pH	Optimal Temp. (°C)	K_m_ (mM)	V_max_	Reference
*Enterococcus avium* M5 ^a^	MRS + 3% MSG	*gadC* *B*	466	4.5	55	3.26	0.012 mM/min	[44]
*Latilactobacillus sakei* A156 ^a^	MRS + 3% MSG	*gadC* *B*	479	5.0	55	16.0	0.011 mM/min	[48]
*Latilactobacillus sakei* OPK 2-59	MRS + 1% MSG	ND ^b^	ND	5.0 ^c^	30 ^c^	ND	ND	[49]
*Levilactobacillus brevis* 877G	MRS + 1% MSG	ND	468	5.2	45	3.6	0.06 mM/min	[53]
*Levilactobacillus brevis* CGMCC 1306	0.17% MSG^c^	ND	468	4.8	48	10.26	8.86 U/mg	[62]
*Levilactobacillus brevis* G144 ^a^	MRS + 3% MSG	*gadC* *B*	479	5.0	40	8.6	0.01 mM/min	[54]
*Levilactobacillus zymae* GU240	MRS + 3% MSG	*gadC* *B*	479	4.5	41	1.7	0.01 mM/min	[59]

^a^ strain isolated from jeotgal. ^b^ ND, not determined. ^c^ crude cell extract was used.

**Table 4 foods-10-02148-t004:** Ornithine production yields by LAB isolated from kimchi.

Strain	Ornithine Yield	Arginine Concentration	Reference
*Levilactobacillus brevis* OPK-3	ND *	MRS + 4% arginine	[71]
*Weissella koreensis* DB1	43.07 g/kg	MRS + 3% arginine	[72]
*Weissella koreensis* OK1-4	27.01 mg/L/h	MRS + 1% arginine	[73]
*Weissella koreensis* OK1-6	31.41 mg/L/h	MRS + 1% arginine	[73]
*Weissella koreensis* SK	7.17 g/L	MRS + 1% arginine	[74]

* ND, not determined.

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
