# Peer review of "Some Important Metabolites Produced by Lactic Acid Bacteria Originated from Kimchi"

_foods, 2021, doi:10.3390/foods10092148_

Round 1

Reviewer 1 Report

The manuscript entitled: “Some Important Metabolites Produced by Lactic Acid Bacteria Originated from Kimchi”reports interesting information on a traditional fermented food part Korean traditional diet. Nonetheless, the end points of the review manuscrit should be cleared better. The Authors refer to “ingredients” used in the kimchi preparation but it seems not clear how lactic acid bacteria are produced. A better Introduction should help as well as avoiding the well known beneficial properties of lactic acid bacteria. Also why Kimchi is “one of the healthiest food in the world”? Please add appropriate references and justify this sentence. The mechanism of the fermentation should be cleared and appropriately referenced in the context of the proposed Review paper.

This food preparation has been analiyzed? Please give the composition and relative amounts of the used starting food matrices. The context should be cleared bette ras well as the discussion. The production of fermented products depends also on the ratio between the ingredients, temperature, pH, etc. Is there a connection between the long and interesting discussion about the kimchi and lactic acid bacteria? These last seem to be the main topic of proposed review paper.

Paragraph 2.2. Please note that GABA is not a nutraceutical: please check the proper recent definition for nutraceutical which are not doof supplements. Besides itis a very long paragraph.

The Conclusion section seems not connected to the Discussion. Please give examples on lactic acid baacteria from Kimchi use in the area of interest. This section should be revised and assess the end points and applications inclusing the perspective point of view of the Authors. Lines 665 and following should be better substantiated by data where available. The main focus of the manuscript is the food preparation or the lactic acid bacteria: this pint should be clearly assessed and substantiated.

Reviewer 2 Report

The manuscript was well structured by the Authors in accordance with the proposed title of the manuscript. The Authors described the selected metabolites produced by kimchi LAB very well. They attempted to discuss the results presented in the available literature and they also made a small meta-analysis especially with regard to the increase in GABA content in kimchi. Besides, the Authors presented possibilities to produce kimchi with better quality, extended shelf-life and improved functionality.

Author Response

Thank you for the kind comments.

Reviewer 3 Report

Thank you for the opportunity to review this article.

The manuscript is extremely well written.

Here are some corrections.

Line 105. You can add the very recent publication

Agriopoulou, S.; Stamatelopoulou, E.; Sachadyn-Król, M.; Varzakas, T. Lactic Acid Bacteria as Antibacterial Agents to Extend the Shelf Life of Fresh and Minimally Processed Fruits and Vegetables: Quality and Safety Aspects. Microorganisms 20208, 952.

Line 120.bacteriocins will...Please correct the the font size

Line 150. fermentation period (6 ℃, 9–12 days...Please correct ℃

Table 3. Please correct ℃

Line 343-344.Please correct the the font size

Author Response

Comments and Suggestions for Authors

Thank you for the opportunity to review this article.

The manuscript is extremely well written.

Response: Thank you for the kind comments.

Here are some corrections.

Line 105. You can add the very recent publication

Agriopoulou, S.; Stamatelopoulou, E.; Sachadyn-Król, M.; Varzakas, T. Lactic Acid Bacteria as Antibacterial Agents to Extend the Shelf Life of Fresh and Minimally Processed Fruits and Vegetables: Quality and Safety Aspects. Microorganisms 20208, 952.

Response: a new reference was inlcuded as suggested, Ref 13

Line 120.bacteriocins will...Please correct the the font size

Response: the font size was corrected. (line 109)

Line 150. fermentation period (6 ℃, 9–12 days...Please correct ℃

Response: line 150, ℃ ,  it was corrected.

Table 3. Please correct ℃

Response: Table 3, ℃ ,  it was corrected.

Line 343-344. Please correct the the font size

Response: font size in line 343-344 was corrected (line 325-326)

Round 2

Reviewer 1 Report

The manuscript has been properly modified and improved. No other modifications seem necessary.